# Silicon in Horticultural Crops: Cross-talk, Signaling, and Tolerance Mechanism under Salinity Stress

**DOI:** 10.3390/plants9040460

**Published:** 2020-04-06

**Authors:** Musa Al Murad, Abdul Latif Khan, Sowbiya Muneer

**Affiliations:** 1Horticulture and Molecular Physiology Lab, School of Agricultural Innovations and Advanced Learning, Vellore Institute of Technology, Tamil Nadu 632014, India; musa.almurad@vit.ac.in; 2School of Biosciences and Technology, Vellore Institute of Technology, Vellore, Tamil Nadu 632014, India; 3Natural & Medical Sciences Research Center, University of Nizwa, Nizwa 616, Oman; abdullatif@unizwa.edu.om

**Keywords:** cross-talk, horticultural crops, oxidative stress, Silicon, salinity stress, signaling pathways

## Abstract

Agricultural land is extensively affected by salinity stress either due to natural phenomena or by agricultural practices. Saline stress possesses two major threats to crop growth: osmotic stress and oxidative stress. The response of these changes is often accompanied by variety of symptoms, such as the decrease in leaf area and internode length and increase in leaf thickness and succulence, abscission of leaves, and necrosis of root and shoot. Salinity also delays the potential physiological activities, such as photosynthesis, transpiration, phytohormonal functions, metabolic pathways, and gene/protein functions. However, crops in response to salinity stress adopt counter cascade mechanisms to tackle salinity stress incursion, whilst continuous exposure to saline stress overcomes the defense mechanism system which results in cell death and compromises the function of essential organelles in crops. To overcome the salinity, a large number of studies have been conducted on silicon (Si); one of the beneficial elements in the Earth’s crust. Si application has been found to mitigate salinity stress and improve plant growth and development, involving signaling transduction pathways of various organelles and other molecular mechanisms. A large number of studies have been conducted on several agricultural crops, whereas limited information is available on horticultural crops. In the present review article, we have summarized the potential role of Si in mitigating salinity stress in horticultural crops and possible mechanism of Si-associated improvements in them. The present review also scrutinizes the need of future research to evaluate the role of Si and gaps to saline stress in horticultural crops for their improvement.

## 1. Introduction

Salinity is a major threat to agriculture under irrigation, thereby instigating damage and inhibition of crop growth and development. More often than not, salinity affects the physicochemical properties of the soil and ecology of the area. Lower agricultural productivity, lower economic returns, and erosions of soil are the eventual consequences of salinity stress [1]. It is estimated that 7% of the land on Earth and 20% of the arable land are salinity affected, and the percentage of salinity affected land is projected to rise to almost 50% by the middle of the 21^st^ century [2,3]. The unsupervised irrigation practices, fertilizer usage, low precipitation, higher surface evaporation, weathering of native rocks, and industrial pollution are the various reasons that can be associated with the emergence of salinity-affected lands [4,5]. The typical definition of saline soil is based on the electrical conductivity (EC) measurements of the saturation extract (EC_e_) in the root zone of a plant. When EC of the EC_e_ in root zone exceeds 4 dS m^−1^ at 25 °C with exchangeable sodium of 15%, the soil is said to be saline. At this EC_e,_ most of the crops are affected, while lower EC_e_s can also be unfavorable to certain crops [3,6]. Thus, strategies to develop tolerant varieties of plants to abiotic stresses are a major challenge so as to surpass the depleting food production system and meeting the global food demands [7].

Salinity stress occurs due to the accumulation of Na^+^ and Cl^−^ ions in the soil at concentrations higher than adequate levels [8]. The overall developmental stages of the plants, viz., germination, vegetative growth, and the reproductive stages, are affected by salinity. These effects can be attributed to the multifaceted interplay of morphological, physiological, and biochemical processes in plants [9,10]. In plants, during salinity stress, the reduction in leaf area, chlorophyll content, and stomatal conductance, directly affects rate of photosynthesis. In addition to this, decreased photosystem II efficiency can also be held accountable for disturbing photosynthesis [11]. Salinity also inhibits microsporogenesis and stamen filament elongation, which leads to programmed cell death, abortion of ovule, and senescence of fertilized embryos, ultimately affecting the reproductive development [12]. Moreover, the deterrent effect of salinity in plants can be seen in the form of osmotic stress and ion-specific stress. The former is known to dismantle homeostasis in water potential due to the accumulation of solutes at higher concentrations than is required, and the latter leads to accumulation of Na^+^ and Cl^−^ in excess quantities, in turn affecting the K^+^/Na^+^ ratio [13]. K^+^ plays a pivotal role in synthesis of proteins, maintenance of turgor pressure in cell, and stimulation of photosynthesis; therefore, its absence can be detrimental for the plants. The osmotic and ionic stresses along with nutrient deficiencies (e.g., N, Ca, K, P, Fe, and Zn) lead to the development of oxidative stress in plants [14]. The reactive oxygen species (ROS) which accumulate in plants have negative effects on cell structures and various other molecules such as DNA, lipids, proteins, and genes [15,16,17]. The mechanisms in plants that regulate the K^+^/Na^+^ homeostasis, concentrations of nutrients (N, Ca, K, P, Fe, and Zn), and balanced ROS production and detoxification can be depended upon to provide tolerance towards salinity stress [18].

Attempts have been made for sustainable management of salinity stress, such as by changing farming systems so as to include perennials in rotation with annual crops (phase farming), in mixed plantings (alley farming, intercropping), or in site-specific plantings (precision farming) [19]. But the execution is restricted by factors such as cost, abundance of water of good quality, or good water resource. Other alternative approaches to mitigate adverse effects of salinity stress include development of salt-tolerant crops and transgenics, application of plant growth-promoting bacteria, endophytes, salt leaching from root zone, and using the drip or micro-jet irrigation technique such that the use of water is optimized [20,21]. However, very little information is available to us regarding the mineral status and plant dynamics towards tolerance to salinity stress [22]. Therefore, a major challenge in this regard is the development of efficient, affordable, and easily adaptable methods for salinity stress management.

Silicon (Si), despite being touted as the second most abundant element in Earth’s crust, its status in terms of its essentiality for growth and development in plants is often debated. This is because of the fact that different plant species have different abilities of Si uptake [23]. On the basis of the mode of Si uptake, plants are categorized into high, intermediate, and non-Si accumulators [24]. Monocots such as *Zea mays* and *Oryza sativa* are known to accumulate silicon in the order of 5% or higher in their tissues on a dry weight basis and are called Si accumulators, whereas dicots, such as *Helianthus annuus* (sunflower) and *Benincasa hispida* (wax gourd), accumulate 0.1% Si in their tissue on a dry weight basis and are commonly called an intermediate type of Si accumulator [25,26]. However, even the plants that are non-accumulators of Si show significant result in mitigating abiotic stress if the nutrient solution or soil is enhanced with exogenous Si [24]. Scientists usually consider Si a ‘non-essential’ element for the development of plants, but some researchers prefer to refer to Si as a ‘quasi-essential’ element in plants of higher orders due to the findings that Si-supplemented plants show better growth than non-supplemented ones [27,28]. Silicon has proved to be very important in alleviating various environmental stresses such as biotic stress (diseases of plants and damage by pests) and abiotic stresses, namely salinity stress, drought stress, freezing stress, and toxicity by heavy metals [29,30,31,32].

Abiotic stresses in general cause numerous alterations in the physiological, molecular, and biochemical processes that operate in plants. In response to abiotic stress, various signaling pathways are activated which results in the development of a multifaceted regulatory network that involves transcription factors, ion homeostasis, antioxidants, hormones, kinase cascades, ROS, and osmolytes synthesis [33,34]. The current times have witnessed the advancement in “omics” technologies that have profound application in plant sciences to identify key proteins or metabolites which are responsible for stress tolerance in plants and the genes that regulate such molecules [35]. Amongst the various “omics” approaches available, proteomics is useful for determination, protein identification, expression profile, post-translational modifications (PTM), and protein–protein interactions in both stress and non-stress conditions [36,37]. A response towards abiotic stress at the molecular level usually involves the change in protein expression pattern, thus making the proteomic approach very suitable in deciphering a link between accumulation of protein during stress conditions and its relation with stress tolerance [38]. Identification of the possible candidate genes by plant stress proteomics can be helpful to genetically enhance the plants against abiotic stresses [39].

The effect of silicon in alleviating the salt-induced injury has been studied in several crops such as wheat, barley, maize, rice, tomato, cucumber, and alfalfa [40,41,42,43]. However, these studies have basically dealt with the morphological and physiological responses of plant towards salinity stress and no, or very little, light have been shed on the proteome response of plants subjected to salinity stress under silicon supplementation. Plant response to salt stress through a proteomics approach has been studied on certain agricultural and horticultural crops, such as durum wheat [44], canola [45], sugar beet [46], soybean [47], peanut [48], sorghum [49], tomato [50], potato [51], and cucumber [52].

Horticultural crops, particularly fruits and vegetables, acquire an important place in the food industry. Horticultural crops have gained much importance in recent years because, besides their human consumption, they play an important role in commerce like export trade and processing industries. Horticultural crops also produce employment to the farm population, transport, processing industries, and self-seeking employment in the form of entrepreneurs. However, due to several abiotic and biotic stresses, horticultural crop production is drastically decreasing. Keeping this view into consideration, it is important to emphasize production of horticultural crops in global plans and how to improve their production should be taken into consideration seriously. To improve the production of agronomical crops with Si supplementation, a lot of research has been carried out. However, very limited studies have been carried out to elucidate the effect of Silicon in mitigating salinity stress in horticultural crops employing proteomic approaches. Since the proteomics approach deals with the identification and characterization of stress inducible proteins, detailed research on the efficiency of silicon to alleviate salinity stress employing a proteomic approach can help us to comprehensively illustrate the process of stress tolerance in plants induced by Si.

In the first half of the review, we discuss the plant cellular mechanisms that are involved in salinity stress tolerance at the physiological and biochemical level, such as ROS production and detoxification, role of ion pumps, phytohormones, transcription factors, osmoprotectant, etc. In the second half of the review, we focus on cross-talk, signaling pathways, and tolerance mechanism of plants towards salinity stress by elucidating the interaction between silicon and salinity stress in horticultural crops employing a proteomic approach. The current review will gather the available information and decipher the role of Si in alleviating salt stress in the field of horticulture by proteomic approach. The review will help researchers to apprehend the gaps in the understanding of the mechanism involved in salinity stress tolerance in crops that are either consumed on daily basis in the form of vegetables and fruits or commercially applicable in the form of flowers. 

## 2. Reactive Oxygen Species (ROS) and their Production under Salinity Stress

In any given time in a plant cell, the rate of production of ROS and its quenching is at equilibrium. However, this equilibrium is disturbed when plants are affected by abiotic stresses, such as salinity stress, heat stress, chilling stress, drought stress, etc., and ROS is overproduced. The ROS production center in plants is organelles, such as chloroplasts, mitochondria, and peroxisomes.

### 2.1. In Chloroplasts

The PSI of the thylakoid in chloroplasts is the major producer of superoxide, whereas the PSII is involved in the production of singlet oxygen (O_2_^−1^). In PSI, superoxide is formed by the reduction of a single electron of molecular oxygen by plastosemiquinone in the plastoquinone pool and also by the involvement of ferredoxin (Fd) and/or iron sulfur redox centers in the electron transport chain (ETC) [53]. The rapid production of hydrogen peroxide from superoxide is generally found to be spontaneous or due to SOD (superoxide dismutase). In the PSII reaction center, the excited triplet chlorophyll molecules are known to excite the oxygen (^3^O_2_) in its triplet ground state to an excited single state (^1^O_2_) [54,55]. Salt stress is known to cause the stomata to close, thus disturbing the CO_2_-to-O_2_ ratio in leaves and inhibiting photosynthesis. Such conditions in plants lead the electrons to leach out oxygen, thus increasing the rate of formation of ROS [56]. 

### 2.2. In Mitochondria

The complex I (NADH ubiquinone oxidoreductase) and complex II (ubiquinol-cytochrome c oxidoreductase) of the mitochondrial electron transport chain is majorly responsible for the production of O_2_^−^ [57,58,59]. The dismutation of O_2_^−^ that was produced from the electron transport chain leads to generation of H_2_O_2_ and O_2_. Being considered a low toxic compound, H_2_O_2_ can react with Fe^2+^ and Cu^+^ to form highly toxic compounds such as hydroxyl radicals and can diffuse out of mitochondria to other cellular compartments [60].

### 2.3. In Peroxisomes

The peroxisomal matrix and the peroxisomal membranes are the two sites known for O_2_^−^ production in the peroxisomes. In the peroxisomal matrix, the oxidation of xanthine and hypoxanthine is catalyzed by xanthine oxidase to produce uric acid and further leads to production of O_2_^−^ radicals [61]. In the peroxisomal membranes, three peroxisomal membrane polypeptides (PMPs) of the electron transport chain in peroxisomes are involved in the generation of O_2_. These PMPs have been characterized to have a weight of 18, 29, and 32 KDa [62]. H_2_O_2_ is produced in the peroxisomes via a direct pathway or via disproportionation of O_2_^−^ generated in the peroxisome. Glycolate oxidase catalyzes glycolate in the process of photorespiration, thus generating H_2_O_2_. The β-oxidation of fatty acids is also known for generating H_2_O_2_ [63]. Lipid peroxidation of peroxisomes is enhanced by salt stress.

The electron transport chain constituted by the flavocytochrome unit of the NAPDH oxidase in plasma membrane is involved in the reduction of O_2_ to O_2_^−^. The NADPH oxidase is the enzyme postulated to be responsible for ROS in salt stress [64]. Amine oxidases, germin-like oxalate oxidases, and cell wall peroxidases that are pH dependent are also assumed to be involved in the ROS production and accumulation in apoplast [65,66,67]. The interplay of the various organelles involved in the production of ROS during salinity stress conditions is shown in Figure 1.

## 3. ROS Detoxification: A Response towards Salt Stress Tolerance in Plants

### 3.1. In Chloroplasts

The first line of defense against ROS starts in thylakoid with the formation of the thylakoidal scavenging system, the major constituents of which are thylakoid superoxide dismutase/SOD (tSOD), thylakoid ascorbate peroxidase/APX(tAPX) and Ferredoxin-dependent reduction of mondehydroascorbate (MDA) [68]. The in-situ disproportionation of O_2_**^−^** (that is generated in PSI) into H_2_O_2_ and O_2_**^−^** occurs by a reaction catalyzed by tSOD. The H_2_O_2_ thus produced is then reduced by ascorbic acid (AsA) to water with the catalytic action of tAPX, and oxidation of AsA into MDA radical occurs. The reactions above constitute the water–water cycle. Reduced ferredoxin is known for the reduction of MDA to AsA. The MDA sometimes also disproportionate to dehydroascorbate (DHA). Further, the ascorbate glutathione cycle (AsA–GSH) reduces MDA or DHA to AsA [69]. Detoxification of the ROS that escaped from the thylakoid or grana is carried out by stromal SOD, stromal APX (ascorbate peroxidase), and AsA–GSH cycle of the stroma. H_2_O_2_ removal in stroma is also carried out by PrxR and APX cycle [68].

The ^1^O_2_ that is produced in the chloroplasts reacts with other molecules causing oxidative damage to proteins, lipids, and DNA. This ^1^O_2_ generated is quenched by two molecules of β-Carotene in PSII. ^1^O_2_ can be scavenged by tocopherol as well but at a rate two folds lower than that of β-Carotene [70]. AsA and GSH, being non enzymatic antioxidants, are the central cellular redox buffer. As the concentration of AsA is high in chloroplasts, it can effectively scavenge ROS such as superoxide, hydroxyl radicals and singlet oxygen [71]. However, the enzymatic scavenging system is precedented by APX, SOD (superoxide dismutase), and GR (glutathione reductase) of the chloroplast. Previous studies [72,73] showed that SOD, APX, and GR that are thylakoid and stromal bound, in halophyte chloroplast *Suaeda salsa* L., are enhanced when subjected to salinity stress, which can be regarded as an essential mechanism in halophytes for mitigating salinity stress. The overexpression of cytosolic APX in chloroplasts of plants is found to improve tolerance towards salinity and drought [74]. The expression of SOD and APX in chloroplasts of transgenic tobacco plants was also found to improve the tolerance against oxidative stress mediated by MV [75]. These results ascertain the fact that chloroplasts play a major role in protection against oxidative damage that is induced by external forces. 

### 3.2. In Mitochondria

The O_2_^−^ generated in mitochondrial electron transport chain is scavenged by two mechanisms. In the first, O_2_^−^ is converted to H_2_O_2_ by the catalyzing enzyme Mn-SOD [68,76,77]. In the second, spontaneous dismutation converts O_2_^−^ into H_2_O_2_. The H_2_O_2_ is then acted upon by mitochondrial APX and removed through the AsA-GSH cycle [78,79]. Peroxiredoxins (Prxs) use thioredoxins as a reductant source to reduce the H_2_O_2_ produced. Thioredoxins can later be reduced by thioredoxin reductase [80]. Rather than scavenging the O_2_^−^ produced, mitochondria can instead modulate the production of O_2_^−^ itself via two mechanisms. The first mechanism involves the maintenance of a basal ubiquinone pool reduction state by alternative oxidase (AOX), reducing the production of mtROS [81]. In the second mechanism, a proton leak occurs across membrane due to the uncoupling of an uncoupling protein (UCP), thus removing the inhibition of mtETC and consequently decreasing the mtROS production [82].

It has been found that the wild tomato species *Lycopersicon pennellii* tolerant to salt shows an upregulation in the levels of ASA and GSH and SOD activity [83]. Mitochondrial MnSOD at the transcript level were shown to be induced by salt treatment in the variety tolerant to salt but not in the variety sensitive to salt [84]. In another study, it was reported that mitochondrial MnSOD of *Nicotiana*
*plumbagnifolia* when overexpressed in *Nicotiana tabacum* mitochondria, protected the latter from instances of oxidative damage [85]. 

### 3.3. In Peroxisomes

Three out of nine plant peroxisomal SOD, CuZn-SOD, and Mn-SOD from watermelon and Mn-SOD from pea leaves were purified and characterized [86]. Their function is to convert O_2_^−^ produced in the peroxisomes into H_2_O_2_ and O_2_. The H_2_O_2_ is further acted upon by CAT (catalase) and Asa-GSH cycle and converted to H_2_O [68]. The peroxisome matrix contains Dehydroascorbate reductase (DHR) and GR, whereas the peroxisomal membrane contains APX and mondehydroascorbate reductase (MDAR) [86]. MDAR, in order to provide a continuous supply of NAD+ essential for peroxisomal metabolism, re-oxidizes NADH. The antioxidant enzymes bound to membrane prevent the leakage of H_2_0_2_ from peroxisomes [87]. It has been reported that leaf peroxisomes of tomato plants contain GPX [88], and peroxisomal matrix of pea leaves are localized with Prx [89], both of which are known to decrease H_2_O_2_ levels. Salinity stress causes upregulation of the levels of ASA and GSH, as well as SOD, APX, CAT, and MDAR activities in root peroxisomes of the tomato species *Lycopersicon pennellii* that is tolerant to salt [83]. Therefore, a comprehensive knowledge of the mechanisms involved in ROS detoxification in plants under salinity stress can provide new insights in plant stress tolerance. 

## 4. Role of Ion Pumps, Calcium, and SOS Pathway in Maintaining Ion Homeostasis during Salinity Stress

High salinity stress causes an excess of Na^+^ ion accumulation in plants leading to imbalance in Na^+^ homeostasis. Various components of the cells, such as ion pumps, calcium sensors, and their downstream interacting partners, take part in restoring the ion homeostasis and efflux of Na^+^ from the cells. However, there is bias sometimes in selectivity towards ions between channels. For example, K^+^ inward rectifying channel is known to enhance the K^+^ influx when there is plasma membrane hyperpolarization leading to accumulation of K^+^ ions rather than the usual Na^+^ accumulation. The histidine kinase transporter (HKT) also has a low affinity for Na^+^ ion transport and thus facilitates the blockage of Na^+^ ions into cytosol [90]. There is also presence of a voltage independent nonspecific cation channel, serving as a gateway for the influx of Na^+^ into cells. During the event of plasma membrane depolarization, the K^+^ outward rectifying channel carries out the efflux of K^+^ and facilitates the Na^+^ influx into the cell thus leading to accumulation of Na^+^ in cytosol. Excess Na+ ions are flushed into the vacuoles by the vacuolar Na^+^/H^+^ exchanger (NHX). Generation of an electrochemical gradient by H^+^-ATPase leads to the passive movement of H^+^ along the electrochemical gradient coupled by NHX, and simultaneously Na^+^ ions are pushed out of the cytosol. The Ca^2+^ homeostasis is carried out by the H^+^/Ca2^+^ antiporter (CAX1) pump [91,92,93].

Calcium is also considered to be in the epicenter of regulation of signaling pathways during salinity stress. The condition of salinity stress leads to an increase in the cytosolic Ca^2+^, leading to the activation of signal transduction pathways for tolerance towards salinity stress. Ca^2+^ release is stipulated to be due to two major events [94]. One event involves the activity of EGTA or BAPTA in blocking the calcineurin-mediated activity leading to release of Ca^2+^ ions form an extracellular source (apoplast). The other event involves the Phospholipase C activation; as a result, phosphatidylinositol bisphosphate is hydrolyzed into inositol trisphosphate, leading to release of Ca^2+^ from intracellular Ca^2+^ stores. A level up in calcium signaling is seen due to the presence of calcium sensors which detect the calcium signatures, decode them and process the information downstream leading to phosphorylation cascade initiation and gene expression [94].

The three genes, *SOS1*, *SOS2*, and *SOS3* (salt overlay sensitive) were identified as the result of the work of Wu et al. (1996), who carried out a mutant screen for Arabidopsis plants that were oversensitive to salinity stress. A Ca^2+^ binding protein, known as calcineurin B-like protein (CBL), important in detecting Ca^2+^ concentration in the cytosol and relaying the signal downstream, is encoded by *SOS3* (AtCBL4) [95]. CBL interacting protein kinase (CIPK), which is novel serine/threonine protein kinase, is encoded by *SOS2*. The function of *SOS3* is to activate *SOS2* protein kinase activity in a calcium-dependent manner [91]. Genetic analysis established the fact that *SOS1*, *SOS2*, and *SOS3* function collectively to provide tolerance against salinity stress [96,97]. *SOS1* was directly phosphorylated by the action of *SOS3*-*SOS2* kinase complex. It is ascertained that the SOS pathway opens up various branches that are actively involved in sequestration of Na^+^ out of the cells for maintaining ion homeostasis. NHK is also activated by *SOS2* which leads to the pushing out of the excess Na^+^ ions into the vacuoles, again maintaining ion homeostasis. Calnexin and calmodulin, which are the calcium binding proteins, are known to detect the calcium concentration, thus activating NHK. An additional target for *SOS2* has been identified to be the *CAX1* conferring homeostasis of Ca^2+^ in the cytosol [94]. A schematic representation of SOS and a related pathway involved in maintenance of ion homeostasis is shown in Figure 2.

## 5. Role of Phytohormones and Transcription Factors during Salinity Stress

The role of abscisic acid (ABA) as a phytohormone in regulating growth and development of plant and also its response towards salinity stress are well documented [96,98]. Stress conditions lead to the activation of genes for enzymes carrying out ABA biosynthesis consequently increasing the levels of ABA. A phosphorylation pathway that is calcium dependent regulates various ABA biosynthetic genes such as zeaxanthin oxidase, 9-cis-epoxycarotenoid dioxygenase, ABA-aldehyde oxidase, and molybdenum cofactor sulfurase when hit by salinity stress [97,98,99]. When ABA accumulates in excess, it can signal the ABA biosynthetic genes through a calcium dependent pathway to activate the catabolic ABA enzymes to degrade excess ABA. Both ABA-dependent and ABA-independent pathways are responsible for the activation of osmotic stress-responsive genes during salinity stress [100]. Other hormones, namely Brassinosteriod (BR) and Salicylic acid (SA), also have a profound role in response towards plant stress [101]. SA exerts its effects in plant tolerance towards stress by interplay and signaling with various other growth hormones [102]. It is also efficient in overcoming adverse effects of salinity stress [103]. On the other hand, BR application is known to enhance the production of various antioxidant enzymes such as APX, GPX, POX, and SOD and also leads to the accumulation of antioxidants that are non-enzymatic in nature [103].

Cis regulatory elements which are part of the promoters of stress-induced genes such as DRE/CRT, ABRE, MYC, and MYB recognition sequence are acted upon by upstream transcription factors and thus regulated [91,93]. AREB, a leucine zipper transcription factor, when activated by the ABA-dependent salinity stress signaling, binds to ABRE in order to activate genes that are stress responsive (RD29A). The DRE cis element of osmotic stress genes is activated by the transcriptional factors *DREB2A* and *DREB2B* and thus maintains the osmotic equilibrium of the cell [96]. The binding of *MYCRS* and *MYBRS* elements by *MYC/MYB* transcription factors RD22BP1 and *AtMYB2*, respectively, also activates *RD22* gene. Hence, it can be stipulated that these transcription factors have a mechanism to cross talk amongst themselves to provide maximal response during salinity stress. 

## 6. Role of Osmoprotectant Osmolytes during Salinity Stress

The osmoprotectant osmolytes, namely glycine betaine and proline, are synthesized by some plants in response to salinity stress so as to maintain osmotic homeostasis in the cell [98,104]. Choline monooxygenase and betaine aldehyde dehydrogenase synthesize GB (Gibberellin) by their enzymatic action. When the betaine aldehyde decarboxylase-encoding genes from *Suaeda liaotungensis* were overexpressed, the tolerance to salinity stress in tobacco plants was enhanced. Tolerance towards salinity stress in rice was enhanced by the *Arthrobacter globiformis* choline dehydrogenase gene (codA) [104]. In cyanobacteria and Arabidopsis, when the N-methyl transferase gene is overexpressed, it caused the GB to accumulate in levels higher than usual and thus provided tolerance to salinity [96]. In rice [105] and maize [106], exogenous application of GB led to the increase in growth of low or non-accumulating plants under salinity stress. A decrease in the Na^+^ concentration and an increase in K^+^ concentration in shoots were observed when GB was applied to plants affected by salinity stress, as compared to plants that were untreated. Hence, GB can be considered to have a prominent effect in ameliorating salinity stress in plants via transduction pathway and ion homeostasis.

The accumulation of amino acid proline is generally linked with osmotic regulation in cells under salinity stress to emasculate its affects [104]. Stress responsive genes that are known to possess proline response elements (PRE, ACTCAT) in their promoters are activated by proline itself [98]. Pyrroline-5-carboxylate synthetase (P5CS) and pyrroline-5-carboxylate reductase (P5CR) are the two enzymes that are used by glutamic acid for synthesis of proline for plants of higher order [107]. When the gene P5CR was overexpressed in transgenic tobacco, it conferred tolerance towards salinity [108]. Oxidative stress caused by salinity stress causes cell membrane damage. Proline comes into action in this scenario and protects the cell membrane from damage with the help of the antioxidant system [109]. 

## 7. Si Uptake, Transport, and Accumulation

Soil generally contains Si in amounts of about 50 to 400 g Si kg^−1^ [110]. About 50%–70% of soil mass has Si present in it in the form of SiO_2_ and other aluminosilicate forms [23]. Despite the abundance of Si in soil, its uptake by plants is very low due to the poor solubility of Si compounds present in soil [111,112,113]. Factors such as pH of soil, water content, cations, and organic compounds that are present in soil greatly influence the solubilization of Si in soil [114]. The PAFs (plant available forms) of silicon, such as silicic acid or mono silicic acid [Si(OH)_4_ or H_4_SiO_4_], are generally taken up by plants. The range of PAF-Si concentration in soil varies from 10 ppm to 100 ppm [115]. At physiological pH, Si in the form of silicic acid or mono silicic acid can cross the plasma membrane [116]. In soil solution, silicic acid is found at concentrations of 0.1 to 0.6 mM at a pH below 9 [117]. Si concentrations in leaves range from 0.1% to 10% on the basis of dry weight [118,119]. The variation in concentration of Si among species is found to be higher than variation within species [120]. The reason behind variations in accumulation of Si in plants lies in the differential abilities of Si uptake by roots [121]. *Byrophyta*, *Lycopsida*, and *Equisetopsids* are found to accumulate Si in higher concentrations, whereas Filicopsida, gymnosperms, and angiosperms accumulate Si in very low concentrations [26,122]. The cyperaceae, poaceae, and balsaminaceae, which are taxa of angiosperms, accumulate > 4% of Si; 2%–4% of Si accumulation is seen in cucurbitales, urticales, and commenlinaceae, but Solanaceae and Fabaceae are considered to be Si excluders [26,122]. On the basis of water uptake capacity in higher plants, the adsorption of Si at lateral roots can be segregated in three ways: active uptake (uptake of Si is faster than uptake of water). Passive uptake (rate of water uptake and Si uptake is the same) and rejective uptake (uptake of Si is slow as compared to uptake of water) [123,124]. The apoplastic and symplastic routes are used for the uptake of silicic acid by root. NIPs (Nod26-like intrinsic protein), which are a class of the aquaporin (AQP) gene family, are responsible for the symplastic route of Si uptake by roots. Several monocots and dicots have Si transporter AQPs present in them [125,126,127,128,129]. The horsetail, which is a primitive plant species, is considered to be the “king of Si accumulators” [130,131].

Ma et al. [125,132] identified two different Si transporters in rice mutants, namely *OsLsi1* (Si-transporter AQPs, influx) and *OsLsi2* (efflux Si-transporters), responsible for Si uptake. *OsLsi1* and *OsLsi6* (*OsLsi1* homolog), which are influx Si transporter AQPs, carry out the transport of Si between the apoplast and plant cell via the plasma membrane. Thus, the Si influx from soil to root cells is facilitated by the *OsLsi1* gene belonging to NIP-III subfamily of aquaporin [125]. The efflux transporters (*OsLsi2*) carry out Si release into the apoplast (xylem loading), after which the Si is translocated to shoots with the help of a transpiration stream. Thus, the transport of Si from root cells towards stele is facilitated by *OsLsi2* gene (efflux Si-transporter), which is an anion channel transporter [132,133]. Xylem unloading is another crucial event in Si transport so as to prevent Si deposition in xylem. This is facilitated by *OsLsi6*, which is an influx transporter and carries out the Si unloading from xylem into xylem parenchymal cells [133]. Another efflux Si transporter (*OsLsi3)* is stipulated to carry out the reloading of Si into the vascular bundles [134]. Such mechanisms have been observed in plants such as maize and barley [127,128]. The schematic diagram of Si transport in rice is shown in Figure 3.

## 8. Si-Mediated Regulation of ROS

The production of excess ROS in the form of hydrogen peroxide (H_2_O_2_), superoxide (O_2_^−^), and hydroxyl radical (OH) in an amount more than what the plant requires is seen as an immediate consequence of salinity stress in plants, leading to oxidative damage to organelles and membranes [135]. In such a situation, the oxidative stress is mitigated by Si with the production of both enzymatic and non-enzymatic antioxidants of the likes of superoxide dismutase (SOD), catalase (CAT), peroxidases (POD), ascorbate peroxidase (APx), ascorbate (AA), and glutathione (GSH) [136]. Studies showed that Si, by regulating the activities of both enzymatic and non-enzymatic antioxidants can provide an effective ROS scavenging system in plants, and the effect would vary with varying plant species.

Lipid peroxidation mediated by the production of ROS is found to be disastrous for the living organisms [137]. In okra roots that were under salinity stress (7 days), foliar application of Si has been found to increase SOD, POD, and CAT activities leading to a decrease in lipid peroxidation; however, varied response were noted across genotypes [138]. Wang et al. [139], in an experiment in alfalfa under salt stress, reported that Si application increased APX activity in roots, shoots, and leaves and POD and CAT activity in shoots and leaves, respectively. A proportionate decrease in oxidative stress due to augmentation in antioxidant enzyme activities in tomato plant under salinity stress in solution [140] as well as sand [141] has also been reported. Similarly, in rice plant under stress, Kim et al. [142] reported that supplementation of Si reduced the oxidative stress by increasing the antioxidant enzyme activity. 

A study by Liang et al. [32] showed that Si can increase CAT, SOD, and GR in barley but not the APX activity. However, the exogenous application of Si in cucumber leads to an increase in SOD, APX, GR, and GPX activities but had zero effect on the activity of CAT [24]. The accumulation of H_2_O_2_ in sorghum is reduced by the foliar application of Si, leading to an increase in water uptake, which is otherwise affected by the hindrance of excess H_2_O_2_ in the aquaporin activity [143]. Similar results were obtained in plants like tomato [115], grapes [144], wheat [145], okra [138], and rice [146]. In *Glycyrrhiza uralensis,* the POD activity was found to increase on the application of Si in 1, 2, 4, and 6 mM of concentration. In the same study, it was observed that 4 mM of Si increased the SOD activity, whereas the MDA content (malondialdehyde) was decreased in all concentrations of Si as compared to control [147]. Garg and Bhandari [148] reported that the *Cicer arietinum* genotypes exposed to prolonged salinity, showed the presence of oxidative markers such as O_2_^−^, H_2_O_2_, and MDA, the levels of which declined on the application of 4 mM Si.

The above studies decipher the role of Si as an ROS scavenging system by increasing the activities of certain antioxidant enzymes thereby countering the detrimental effects of lipid peroxidation. However, the responses vary from plant to plant, and a clear understanding of the interplay between Si and its interaction with various antioxidant enzymes is poorly explored. Researchers have to come up with unified mechanisms explaining the factors that lead to variations in response of plants in antioxidant enzyme production during Si supplementation to salinity stressed plants. Moreover, the results from a hydroponics setup cannot be apprehended to the result obtained in soil. Further research must link the results obtained from hydroponics to those obtained in soil, such that a unified understanding of the shortcomings and advantages of each of the medium is known and a suitable medium (hydroponics or soil) can be preferred so as to get the best results of Si supplementation under salinity stress. Research on organ specific proteomics has to be carried out to elucidate which organ (leaf to root) is involved in the signal transduction so as to carry out ROS scavenging. 

## 9. Si-Mediated Na^+^ and K^+^ Homeostasis

When plants are under salinity stress, a condition occurs where there are increased Na^+^ and Cl^−^ levels and a simultaneous decrease in K^+^ and Ca^2+^ levels [149,150,151]. Such a condition affects cellular metabolism, causing retardation in plant growth and production of ROS in excess [96]. Tuna et al. [145] reported that Si application in salt stressed plants could reduce Na^+^ uptake and maintain K^+^/Na^+^ levels. In a study by Abbas et al. [138], it was found that application of Si in okra (*Abelmoschus esculentus* L.) affected by salt stress led to a decrease in Na^+^ and Cl^−^ in shoots and roots in addition to which relative water content also increased. In salt-stressed cultivars of Egyptian clover, diatomite Si application led to a dose-dependent decrease in Na^+^ content [152]. Similarly, a decrease in the level of Na^+^ and Cl^−^ was observed in roots of salt stressed grapevine (*Vitis vinifera* L.) upon application of Si [144]. An improved homeostasis of Na^+^/K^+^ as a result of the decrease in Na^+^ and increase in K^+^ uptake was observed in salt-stressed wheat, when Si in the concentrations of 50–200 ppm were applied, with better effects at higher concentrations [153]. In conclusion, SARC-5 (genotype that is salt tolerant) was reported to be more efficient than Auqab-200 (genotype that is alt sensitive). Ali et al. [154] reported similar results on wheat genotypes in saline field on application of Si. Gurmani et al. [155] also observed a decrease in transport of Na^+^ and increased Na^+^/K^+^ ratio upon application of Si in wheat genotypes under salt stress.

Silicon is known to reduce the accumulation of Na^+^ in roots/shoots. Liang and Ding [156] reported an even distribution of Na^+^ and Cl^−^ in the whole root section of barley under salt stress on application of Si and also a decrease in Na^+^ and Cl^−^ levels but increase in K^+^ levels, which is considered as a profound mechanism involved in Si-mediated tolerance against salt stress. Similarly, it has been observed that in alfalfa (*Medicago sativa* L.), Si application led to decrease in Na^+^ level in roots rather than shoots and an increase in K^+^ level in shoots [151]. Whereas Gong et al. [157] reported a significant reduction of Na^+^ in shoots of rice but not roots, on application of Si, thus correlating with the fact that silicon improves shoot growth. Application of Si led to a decrease in leaf apoplsat Na^+^ level, compared to Si-untreated faba bean [158]. These results show that Si mediates salt stress tolerance via distributing Na^+^ to different parts of plants. 

Reduction of relative water content (RWC) of leaf generally serves as a marker for osmotic stress [159]. Plants such as wheat [160], tomato [141], maize [161], sorghum [162], and turf grass [163] which are under salt stress showed an improvement in RWC on application of Si. In wheat leaves, osmotic potential dropped on application of Si [164]. Therefore, mediation of osmotic potential by Si may be considered as an effective mechanism of regulating salt stress. However, further studies are needed to study routes through which Si facilitates water movement under salt stress.

The Na^+^/H^+^ antiporter comes into play in this scenario as it removes Na^+^ from the cytosol into the vacuoles, thus maintaining Na^+^ at low concentrations [165]. From Arabidopsis, *SOS1* gene, encoding the plasma membrane Na^+^/H^+^ antiporter, has been cloned [166]. Compartmentation of Na^+^ is carried out by Na^+^/H^+^ antiporter, driven by H^+^-ATPase and H^+^-pyrophosphatase (H^+^- PPase) [166]. Salt-stressed barley roots normally showed a reduced activity of membrane-bound H^+^-ATPase. However, upon addition of Si, the H^+^-ATPase activity increased considerably, which led to Na^+^ efflux from cell [40]. On addition of Si in root tonoplast of salt stressed barley, Liang et al. [167] reported an increase in H^+^-ATPase and H^+^-PPase activity, which is supposed to enhance the Na^+^ exit into the vacuoles with the help of Na^+^/H^+^ antiporter. Silicon was also reported to increase uptake of K^+^ via enhancing the activity of H^+^-ATPase in soil and hydroponic conditions [168]. This suggests that Silicon encourages plasma membrane H^+^-ATPase activity and tonoplast H^+^-PPase activity and therefore reduces the level of Na^+^ and increases the K^+^ level in cytoplasm. However, the role of silicon in the regulation/expression of Na^+^/H^+^ antiporter during salinity stress is yet to be studied in detail. 

## 10. Si-Mediated Biosynthesis of Compatible Solutes and Phytohormone

In addition to producing antioxidants, under stress conditions, plants also respond to stress by the accumulation of compatible solutes such as proline [169], glycine betaine [170], polyols [171], and carbohydrates [172]. The compatible solutes are hydrophilic in nature and are known to accumulate in high concentration without any disturbance to the biochemical reactions going on in the cell [92]. The high ion concentrations disturb the enzyme activity; such a condition is alleviated by compatible solutes which function to stabilize proteins and its complexes and/or membranes that are under stress [173]. Oxygen radical scavenging is also touted to be the function of compatible solutes [174]. In a study conducted by Seckin et al. [175], it was found that in the roots of wheat that are sensitive to salt, an addition of mannitol increased the antioxidant enzyme activity, thus preventing any sort of oxidative damage due to salt stress. Osmotic protection and tolerance towards salt is synonymous to proline application in higher plants [176]. In leaves of *Populus euphratica*, NaCl stress and mannitol led to the accumulation of proline [177]. However, studies conducted in several plants such as soybean [178], wheat [145], barley [179], sorghum [162], and grapevine [144] reported that the addition of Si resulted in lowering of proline level. Yin et al. [162] reported that application of Si in salt-stressed sorghum for a short period of time led to an increase in sucrose and fructose levels, consolidating that Si is effective in alleviating osmotic stress induced by salt. Since the role of proline is not clearly explained by the previous research, wherein some researchers say proline accumulation causes alleviation of salinity stress, whereas other researchers state that lowering of proline levels alleviates stress under Si supplementation. Therefore, research has to be done to understand the relation of Si application and biosynthesis of compatible solutes and its role in stress tolerance under Si supplementation. 

High concentration of salt has been known to bring about changes in level of plant growth hormones [171]. Under osmotic stress, abscisic acid (ABA) which is commonly known as “stress hormone” is seen to be upregulated and causes alterations in gene expression enhancing the survival of plants under stress [180]. Karmoker and von Steveninck [181] reported that ABA causes inhibition of Na^+^ and Cl^−^ transport to shoot in bean seedlings. In soybean, ABA level was increased during salt stress, but the addition of Si resulted in a decrease in the levels of ABA [178]. Kim et al. [142] reported that short (6 to 12 h) exposure of salt-stressed rice to Si led to downregulation of Jasmonic acid (JA), but the ABA level was found to increase after 6 and 12 h, and then the ABA level dropped after 24 h. In salt-stressed rice, the expression of genes related to ABA biosynthesis, such as zeaxanthin epoxidase and 9-cis-epoxicarotenoid oxygenase 1 and 4 (ZEP, NCED1, and NCED4), is increased on application of Si [142]. The detrimental effects of NaCl stress can be inhibited by application of gibberellins (GA) exogenously [182]. Seed germination and shoot elongation in plants is also influenced by GAs [159]. Lee et al. [178] found that under salt stress, the GA level decreased, but addition of Si led to an increase in GA levels. It is reported by Kim et al. [183] that the genes involved in JA and ABA biosynthesis are regulated by Si; however, effects are said to be time dependent. However, regulation of phytohormone by silicon under salt stress and drawing a relation between the two is yet to be done. 

## 11. Si Efficiency in Salinity-Stressed Horticultural Crops Employing Proteomic Approaches

The proteomic technique is one the most common and advanced techniques to analyze signaling homeostatic pathways under abiotic stress. Several proteomic researches have been performed in model plants like Arabidopsis, tomato, and rice under Si efficiency and various abiotic stresses. However, limited reports are being reported in horticultural crops under abiotic stresses employing proteomic approaches. Therefore, in the current review, we have summarized the research performed on horticultural crops under abiotic stress (salt stress) and Si efficiency.

Muneer and Jeong [140] analyzed the root proteomics of salt-stressed tomato (*Lycopersicon esculentum* L.) supplemented with Silicon (Na2SiO3). A reduction in protein spots and downregulation of proteins were observed under salinity stress (25 and/or 50 mM NaCl/-Si). However, at higher concentrations of salinity, the loss of proteins was higher, compared to lower concentrations, but silicon replaced this loss effectively. A complete reduction in biosynthesis of metabolites and defense-related proteins is attributed to the downregulation of proteins in salt stressed roots, also affecting the antioxidant and nutrient transport signal transduction pathways [184]. About 40 proteins were found to be downregulated in roots affected by salinity stress (25 and/or 50 mM NaCl/-Si) and the same 40 proteins were seen to be upregulated when Si was supplemented to the salinity stressed roots (25 and/or 50 mM NaCl/-Si). Of the various proteins identified, 17% of the proteins identified were found to be related to stress responses, 11% to plant hormones, 11% to cellular biosynthesis, and the remaining proteins were found to be related to transcriptional regulation, RNA binding, and secondary metabolisms. Stress-related proteins, namely Os02g0282000 protein, COPINE 1 family Protein, and zinc finger A20, AN1 domain-containing stress-associated protein, caffeoyl-CoA O-methyltransferase, NBSLRR disease resistance protein, and pathogenesis-related protein 10 were identified. Proteins related to transcriptional regulation, such as transcription elongation factor and transcription elongation factor SPT4 homolog, were identified. Proteins such as potassium channel AKT2, AKT2/3-like potassium channel, and gibberellin 20-oxidase were found to be related to plant hormones. Abiotic stresses such as salinity stress lead to the downregulation of various types of transcriptional proteins [185,186]. Salinity stress results in disturbance of various signaling pathways due to the reduction of transcriptional proteins, which, however, is seen to increase when Si was added (+NaCl/+Si), and thus the essential pathways were restored for optimum functioning [140]. It is also established by various reports that plant hormones play a regulatory role in development pathways and tolerance towards abiotic stresses such as salt stress [187,188]. Muneer and Jeong [140] showed that plant hormone proteins were downregulated in salinity stress but were upregulated again when Si was added to the salinity-stressed plants, indicating the role of Si in affecting the regulatory role of plant hormones for coping with salinity stress.

In another study by Soundararajan et al. [189] on salt-stressed *Rosa hybrida* ‘Rock fire’, the proteomic analysis revealed the essentiality of Si (K2SiO3) in ameliorating salt stress. Salt stress directly affects the expression of photosynthetic proteins due to decline in development and photosynthetic process impairment [190]. In such a situation, Si gets indulged in essential carbon fixing cycles such as Calvin cycle, tricarboxylic acid (TCA) cycle, and pentose phosphate cycle and causes stimulation of photosynthesis-related proteins. In Si treatments, proteins such as RuBisCo (photosynthetic protein) and Ycf4 were found in higher amounts ensuring photoprotection and physiological development by improving the light harvesting process [189]. Si caused an increase in the expression of enzymes β-glucosidases, β-galactosidases, and glucose-1-phosphate adenylyltransferase large subunit, implying its effect on starch and sucrose metabolism. Similarly, acetyl-CoA carboxylase, a precursor enzyme known to play a major role in fatty acid synthesis, is also increased on Si inclusion. Glycerol-3-phosphate dehydrogenase (GPDH) (NAD+), which usually maintains the NADH yield and redox potential of mitochondria under salt stress, is also increased on Si application. This was in agreement with the study of Muneer and Jeong [140] on *Lycopersicon esculentum* L. Under salinity stress, proteins such as ribosome-recycling factor and tRNA (Ile)-lysidine synthase, associated with amino acid biosynthesis, were downregulated but were enhanced upon Si inclusion. Ubiquitin conjugating enzyme E2 8 and E2 36, which were affected by salinity stress, recovered when Si was applied. This might enhance the Si–ubiquitin interaction leading to improvement in protein regulation during the event of post-translational modifications. Thus, an overall increase in the transcription-related protein can cause an overall enhancement in cellular processes under the influence of Si. 

In another study, Manivannan et al. [191] analyzed the leaf proteome of *Capsicum annuum* ‘Bugwang’ under salinity stress, supplemented by Si (K2SiO3). About 245 protein spots were identified, out of which 129 were expressed differentially. The downregulation of 83 spots and upregulation of 46 spots were observed during salinity stress. Under salinity stress, the decrease in protein expression can be attributed to the reduction of protein synthesis caused by reduction in signal transduction and gene regulation pathways [185]. Moreover, the degradation of proteins can be due to excess ROS production leading to incorrect protein folding/assembly in salt-stressed *Capsicum* [192]. However, inclusion of Si led to the upregulation of 67 protein spots [192]. Proteins such as Adenylosuccinate synthase (involved in purine metabolism) leading to enhanced growth and biomass in capsicum, E3 ubiquitin ligase (responsible for floral development, photo morphogenesis), RuBisCo (carbon fixation), and oxygen evolving enhancer protein (photosynthesis related protein) were enhanced by the addition of Si. Resistance towards plant disease and hormone signaling are taken care of by nucleoporins [193]. A nucleoporin-like protein was found to accumulate on application of Si. The expression of RNA polymerase II transcription subunit 11, ribosomal protein L16, and resistance protein candidate was enhanced on Si application. Several other proteins such as Molybdopterin synthase catalytic subunit (ABA biosynthesis), 𝛽-keto acyl reductase (fatty acid metabolism), reverse transcriptase, and eukaryotic translation initiation factor 3 subunitD were found to be upregulated by salinity. MADS-box transcription factor 26 isoform X2 was upregulated on NaCl and Si in combination. Addition of Si to salt-stressed capsicum also upregulated cullin 1D, which is involved in the ubiquitin-proteasome pathway. Proteins involved in major metabolic processes, such as phosphoglycerate kinase, ATP-synthase CF1𝛼 subunit, disease resistance protein RPS2, and double-stranded RNA binding protein 2, were increased in NaCl+ Si treatment [191]. The list of different proteins conferring different responses to the plants under salinity stress on application of Si, as studied by the abovementioned researchers, are listed in Table 1.

## 12. Conclusions

Horticultural crops are important sources of food for human consumption; they not only improve the health/nutrition but also improve the economy globally. To improve the production of horticultural crops, every country should keep the horticultural industry as a priority to meet food demands. During the past few years, horticultural production has been decreasing due to several abiotic and biotic stresses. Among abiotic stresses, salinity stress is one the major stresses negatively affecting horticultural crop development via water stress, cytotoxicity, and excessive uptake of sodium (Na^+^) and chloride (Cl^−^) ions and nutritional imbalance. Saline stress also typically causes oxidative stress in the form of formation of reactive oxygen species (ROS). On the other hand, Si is a beneficial element and is proven to be effective in mitigating ROS caused by abiotic stress. The signaling pathways and co-relation of salt stress and Si-efficiency has been observed in a number of agronomical crops like rice, wheat, soybean, etc. There is a lack of knowledge about the role of Si in horticultural crops, and a limited amount of research has been conducted on the role of Si in horticultural crops such as cucumber, capsicum, strawberry, melon, and rose under several abiotic stresses including salinity stress. In this review, we summarized the reports on signaling mechanism, cross-talk, and tolerance mechanism of horticultural crops to salt stress and efficiency of Si. Since horticultural crops are everyday food globally, it is required to reveal the importance of Si to mitigate salt stress (global problem) to depict the signaling mechanisms and homeostatic maintenance. We conclude that researchers need to focus more on horticultural crops for their improvement under any abiotic stress, including salt stress, by Si supplementation.

## Figures and Tables

**Figure 1 plants-09-00460-f001:**
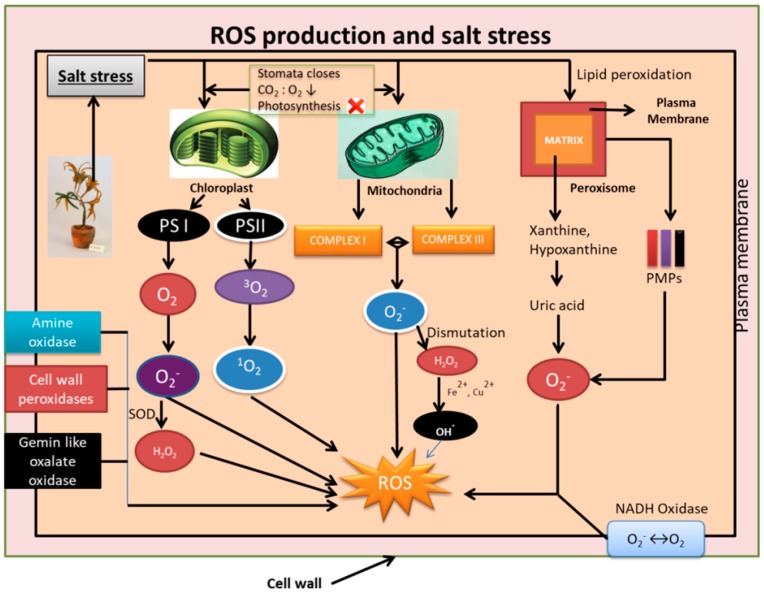
Schematic representation of mechanisms involved in generation of reactive oxygen species (ROS) during salinity stress. Organelles such as chloroplast, mitochondria, and peroxisome are involved in the generation of free radicals such as O_2_^−^, singlet oxygen (^1^O_2_), OH^.^, H_2_O_2_. Apart from this, plasma membrane NADH oxidase, Amine oxidase, cell wall peroxidases, and Gemin-like oxalate oxidases also generate ROS. The generation of ROS is detrimental to cell structures, macro molecules such as DNA, lipids, proteins, and genes affecting the overall functioning of the cell.

**Figure 2 plants-09-00460-f002:**
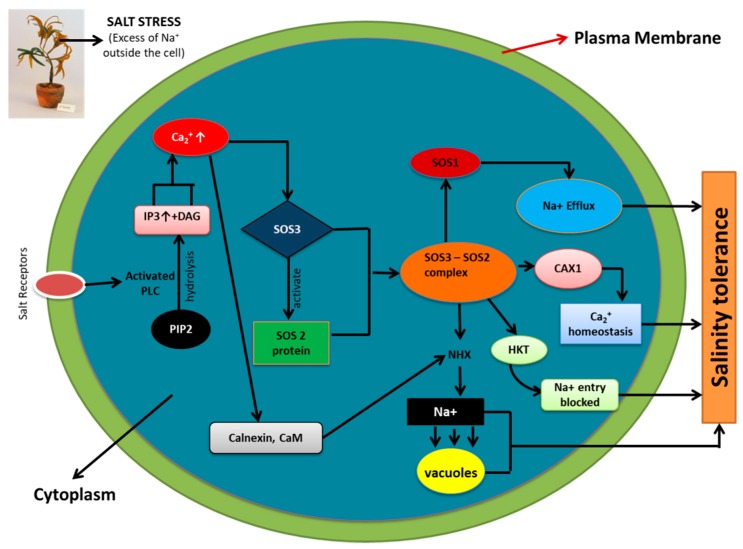
Salt overlay sensitive (SOS) and related pathway involved in maintenance of ion homeostasis (Na^+^, K^+^ and Ca^2+^) during salinity stress. Phosphorylation of SOS1 (Na^+^/H^+^ antiporter) by SOS3-SOS2 protein kinase complex leads to Na^+^ efflux. SOS3-SOS2 complex also inhibits the activity of HKT1 (low affinity Na^+^ transporter), restricting cytosolic entry of Na^+^. The Vacuolar Na^+^/H^+^ exchanger (NHX) is activated by SOS2 and results in Na^+^ sequestration into vacuoles. CAX1 (H^+^/Ca^2+^ antiporter) is involved in Ca^2+^ homeostasis via SOS2 signaling. All these factors, culminatively work together to provide salinity tolerance to plants via maintaining ion homeostasis (Na^+^, K^+^, and Ca^2+^). (Adapted from Tuteja 2007).

**Figure 3 plants-09-00460-f003:**
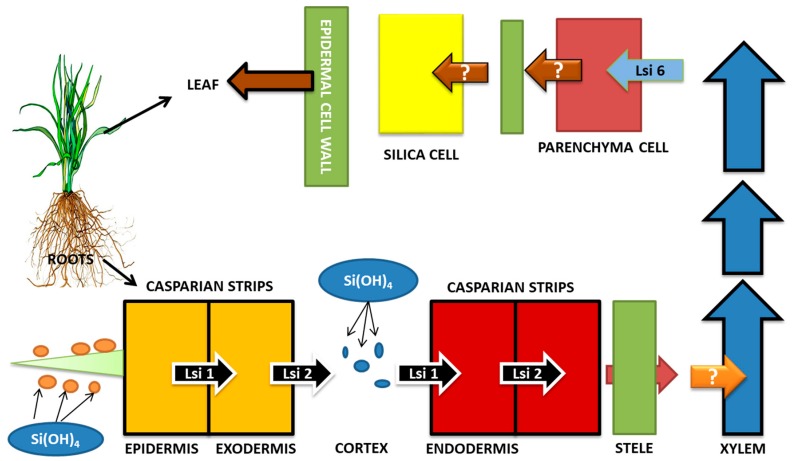
Si transport in Rice (*Oryza sativa*). Influx Si transporters (*OsLsi1* and *OsLsi6*) and efflux Si transporter (*OsLsi2*) are responsible for the transport of Si from soil to roots, then to stele and xylem, and then upwards towards the shoot. Events of xylem loading and unloading and reloading (*OsLsi3*) are also involved. Efficient Si uptake and transport leads to better activity of Si in coping up with salinity stress. (Adapted from Khan et al., 2019).

**Table 1 plants-09-00460-t001:** List of proteins identified in horticultural crops under salinity stress under silicon supply.

Serial No.	Accession Number	Protein Name	Biological Function	Plant Species	Theo./Exp. *pI*	Sequence Coverage (%)	Author
1.	Q6K3C7	Os02g0282000 protein	Defense response	*Lycopersicon esculentum* L.	7.86/3.6	17	Muneer and Jeong (2015)
2.	B9IFL3	COPINE 1 family protein	Defense response	*Lycopersicon esculentum* L.	5.54/5.3	24	Muneer and Jeong (2015)
3.	G7IZ85	Zinc finger A20 and AN1 domain containing stress-associated protein	Stress response	*Lycopersicon esculentum* L.	6.28/5.9	55	Muneer and Jeong (2015)
4.	A2TDB3	Caffeoyl-CoA O-methyltransferase	Stress response	*Lycopersicon esculentum* L.	7.86/3.6	21	Muneer and Jeong (2015)
5.	B0JEM1	NBS-LRR disease resistance protein	Defense response	*Lycopersicon esculentum* L.	6.5/4.6	12	Muneer and Jeong (2015)
6.	Q5DUH6	Pathogenesis-related protein 10	Defense response	*Lycopersicon esculentum* L.	5.1/5.8	21	Muneer and Jeong (2015)
7.	M1C4D6	Transcription elongation factor	Transcriptional regulation	*Lycopersicon esculentum* L.	5.65/5.67	56	Muneer and Jeong (2015)
8.	B9HUZ8	Transcription elongation factor	Transcriptional Regulation	*Lycopersicon esculentum* L.	5.66/6.5	56	Muneer and Jeong (2015)
9.	A9PK54	Transcription elongation factor SPT4 homolog	Transcriptional Regulation	*Lycopersicon esculentum* L.	5.66/5.6	56	Muneer and Jeong (2015)
10.	A9PK54	Transcription elongation factor SPT4 homolog	Transcriptional Regulation	*Lycopersicon esculentum* L.	5.66/5.8	56	Muneer and Jeong (2015)
11.	A9PK54	Transcription elongation factor SPT4 homolog	Transcriptional Regulation	*Lycopersicon esculentum* L.	5.66/6.0	56	Muneer and Jeong (2015)
12.	A9PK54	Transcription elongation factor SPT4 homolog	Transcriptional Regulation	*Lycopersicon esculentum* L.	5.66/6.6	56	Muneer and Jeong (2015)
13.	Q75HP9	Potassium channel AKT2	ABA response	*Lycopersicon esculentum* L.	6.64/7.0	21	Muneer and Jeong (2015)
14.	H9BAN2	AKT2/3-like potassium channel	ABA response	*Lycopersicon esculentum* L.	4.9/4.5	13	Muneer and Jeong (2015)
15.	B2G4V8	Gibberellin 20-oxidase	GA mediated signaling	*Lycopersicon esculentum* L.	5.97/6.5	32	Muneer and Jeong (2015)
16.	P19312	Ribulose bisphosphate carboxylase small chain SSU5B	Photosynthesis	*Rosa hybrida* ‘Rock Fire’	7.60/6.50	28	Soundararajan et al. (2017)
17.	A7M975	Photosystem I assembly protein Ycf4	Photosynthesis	*Rosa hybrida* ‘Rock Fire’	9.59/4.10	28	Soundararajan et al. (2017)
18.	Q7XKV5	β-glucosidase 11	Energy metabolism	*Rosa hybrida* ‘Rock Fire’	7.21/5.90	19	Soundararajan et al. (2017)
19.	Q9SCV4	β -galactosidase 8	Energy metabolism	*Rosa hybrida* ‘Rock Fire’	8.09/5.10	9	Soundararajan et al. (2017))
20.	P12300	Glucose-1-phosphate adenylyltransferase large subunit	Energy metabolism	*Rosa hybrida* ‘Rock Fire’	6.61/6.70	16	Soundararajan et al. (2017)
21.	P85438	Acetyl-CoA carboxylase	Energy metabolism	*Rosa hybrida* ‘Rock Fire’	9.99/4.10	100	Soundararajan et al. (2017)
22.	P85438	Acetyl-CoA carboxylase	Energy metabolism	*Rosa hybrida* ‘Rock Fire’	9.99/5.10	96	Soundararajan et al. (2017)
23.	Q8H2J9	Glycerol-3-phosphate dehydrogenase (NAD+)	Energy metabolism	*Rosa hybrida* ‘Rock Fire’	9.76/6.80	22	Soundararajan et al. (2017)
24	A2YMU2	Ribosome-recycling factor	Transcription/translation	*Rosa hybrida* ‘Rock Fire’	9.35/5.10	23	Soundararajan et al. (2017)
25.	Q32RJ9	tRNA(Ile)-lysidine synthase	Transcription/translation	*Rosa hybrida* ‘Rock Fire’	9.55/5.87	13	Soundararajan et al. (2017)
26.	Q9FZ48	Ubiquitin-conjugating enzyme E2 8	Ubiquitination	*Rosa hybrida* ‘Rock Fire’	6.74/4.47	59	Soundararajan et al. (2017)
27.	P35131	Ubiquitin-conjugating enzyme E2 36	Ubiquitination	*Rosa hybrida* ‘Rock Fire’	6.74/4.60	59	Soundararajan et al. (2017)
28.	XP 004249273	Adenylosuccinate synthetase	Purine metabolism	*Capsicum annuum* ‘Bugwang’	7.5/4.2	25	Manivannan et al. (2016)
29.	XP 008793948	E3 ubiquitin-protein ligase PUB23-like	Photo morphogenesis	*Capsicum annuum* ‘Bugwang’	8.2/4.1	20	Manivannan et al. (2016)
30.	AHL68475	Ribulose-1,5-bisphosphate carboxylase/oxygenase, partial (chloroplast)	Carbon fixation	*Capsicum annuum* ‘Bugwang’	6.7/5.0	41	Manivannan et al. (2016)
31.	XP 009398204	Oxygen-evolving enhancer protein 3-1, chloroplastic-like	Photosynthesis	*Capsicum annuum* ‘Bugwang’	9.5/5.1	52	Manivannan et al. (2016)
32.	XP 003058724	Nucleoporin-like protein	Plant disease and hormone signaling	*Capsicum annuum* ‘Bugwang’	9.1/4.6	33	Manivannan et al. (2016)
33.	XP 004951624	Mediator of RNA polymerase II transcription subunit 11-like	Transcription/translation	*Capsicum annuum* ‘Bugwang’	5.6/6.2	59	Manivannan et al. (2016)
34.	AFB70663	Ribosomal protein L16, partial (chloroplastic)	Transcription/translation	*Capsicum annuum* ‘Bugwang’	11.8/4.7	63	Manivannan et al. (2016)
35.	AAR08850	Resistance protein candidate	Transcription/translation	*Capsicum annuum* ‘Bugwang’	9.4/5.2	100	Manivannan et al. (2016)
36.	XP 010517956	Molybdopterin synthase catalytic subunit-like	ABA synthesis	*Capsicum annuum* ‘Bugwang’	6.5/4.8	80	Manivannan et al. (2016)
37.	AAL83898	Beta-keto acyl reductase	Fatty acid synthesis	*Capsicum annuum* ‘Bugwang’	11.6/6.9	87	Manivannan et al. (2016)
38.	BAB40826	Reverse transcriptase	Transcription/translation	*Capsicum annuum* ‘Bugwang’	7.9/6.3	30	Manivannan et al. (2016)
39.	KHG25806	Eukaryotic translation initiation factor 3 subunit D	Transcription/translation	*Capsicum annuum* ‘Bugwang’	8.9/4.4	26	Manivannan et al. (2016)
40.	XP 008677250	MADS-box transcription factor 26 isoform X2	Transcription/translation	*Capsicum annuum* ‘Bugwang’	8.8/5.4	42	Manivannan et al. (2016)
41.	CAC87838	Cullin 1D	Ubiquitin-proteasome pathway	*Capsicum annuum* ‘Bugwang’	5.0/4.5	25	Manivannan et al. (2016)
42.	KIY92373	Phosphoglycerate kinase, partial	Metabolic processes	*Capsicum annuum* ‘Bugwang’	8.7/4.4	51	Manivannan et al. (2016)
43.	AIF71068	ATP synthase CF1 alpha subunit, partial (chloroplast)	Metabolic processes	*Capsicum annuum* ‘Bugwang’	8.6/5.8	49	Manivannan et al. (2016)
44.	XP 010046336	Disease resistance protein RPS2-like	Metabolic processes	*Capsicum annuum* ‘Bugwang’	5.3/7.0	19	Manivannan et al. (2016)
45.	XP 012064817	Double-stranded RNA-binding protein 2	Metabolic processes	*Capsicum annuum* ‘Bugwang’	8.7/5.7	14	Manivannan et al. (2016)

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
