# Peer review of "Silicon in Horticultural Crops: Cross-talk, Signaling, and Tolerance Mechanism under Salinity Stress"

_plants, 2020, doi:10.3390/plants9040460_

Round 1

Reviewer 1 Report

Authors of the manuscript "Silicon in Horticultural Crops: Cross-talk, Signaling and Tolerance Mechanism under Salinity Stress" summarises studies on the role of silicon in mitigation of salinity stress in horticultural crops as it was stated in the abstract.

My comments are mostly referring to the content and general structure of the manuscript, not to the detailed data as they are usually based on already reviewed literature.

Generally good review paper should not just summarize available literature, but discusses it critically, identifies problems, and points out research gaps. Unfortunately this manuscript misses most of that points and mostly it is report of conducted so far studies.

According to authors statement their work was planned to be focused on role of Si in alleviation of salinity to horticultural crops.

First of all authors do not provide good enough explanation why they are focusing on horticultural crops. In my opinion suggestion that reason is that there is not enough studies suits much better to research study. I would like authors to support their decision by other reasons (physiological, specified susceptibility). Moreover submitted manuscript is not focusing on horticultural crops and their response to SI. Large part of the manuscripts contains description of response of typical crops to salinity.

Some more detailed comments:

L14P1 What is the meaning of "delays" in term of gas exchange?

L6P2- "Salinity stress affects number ..." Sentence not clear should be rewritten.

L27P2 "Monocots such as Zea..." please provide reference

L4p3 "understand the researchers to focus on crops" sentence not clear, please rewrite.

P3-P9 - description is focused on salinity stress, I  suggest to combine it with response to Si

L25P9 "The concentration..." - not informative general statement, obvious. I think the reference [97] is not properly used in the case.

L14-15P11 Word "similarly" do to describes results of both experiments.

L19P11 - style, should be rewritten.

L10P12 should be "apoplast"

L29P13 Do authors consider tomato as horticultural crops? If yes, the sentences should be rewritten.

L31-33P13 Part of the text that should be rewritten. It is not necessary to provide aim of such short paragraph. "under  Si efficiency" - not clear.

L12-13P14 Again very general sentence, please be more precise, focus on salinity.

L18-20P13 The sentence is very similar to earlier sentence on the same study L38P13

L26-28P13 the sentence on the impact of Si on Light harvesting and photoprotection is not supported by cited study [166].

L1P15 Should be [168]

L10P15 RuBisCo was already described above.

Conclusions are poorly written, as I mentioned above, problems and research gaps should be identified. 

I suggest to provide form of Si used in experiments whenever possible it may be useful information.

Author Response

Reviewer #1

Authors of the manuscript "Silicon in Horticultural Crops: Cross-talk, Signaling and Tolerance Mechanism under Salinity Stress" summarises studies on the role of silicon in mitigation of salinity stress in horticultural crops as it was stated in the abstract.

My comments are mostly referring to the content and general structure of the manuscript, not to the detailed data as they are usually based on already reviewed literature.

Generally good review paper should not just summarize available literature, but discusses it critically, identifies problems, and points out research gaps. Unfortunately this manuscript misses most of that point and mostly it is report of conducted so far studies.

According to authors statement their work was planned to be focused on role of Si in alleviation of salinity to horticultural crops.

First of all authors do not provide good enough explanation why they are focusing on horticultural crops. In my opinion suggestion that reason is that there is not enough studies suits much better to research study. I would like authors to support their decision by other reasons (physiological, specified susceptibility). Moreover submitted manuscript is not focusing on horticultural crops and their response to SI. Large part of the manuscripts contains description of response of typical crops to salinity.

Response>

Authors have revised the manuscript as suggested by the reviewer. Authors in brief revised the manuscript to give more structure to the content such as why horticultural crops, gaps in literatures, critical problems and research gaps as commented by the reviewer. Besides, at the end of every sub-heading we have briefly concluded the whole text.

In Brief, authors have revised the Introduction all over again to provide layered information about the magnitude of the problem of salinity (highlighted in red text), the effects it has on plants (highlighted in red text), and the challenges in salinity stress tolerance (highlighted in red text), The subsequent paragraphs introduce Silicon as a saviour in alleviating salinity stress and how proteomics approach (highlighted in red text), of studying salinity stress with Si supplementation can help in identifying stress inducible proteins and its application in genetic engineering for plant tolerance towards salinity stress.

Some more detailed co

L14P1 what is the meaning of "delays" in term of gas exchange?  

Response> 

The term “delays” refers to the consequences of salinity in the form of reduction in leaf area, chlorophyll content and stomatal conductance, ultimately affecting the gas exchange, which in turn affects photosynthesis.

L6P2- "Salinity stress affects number ..." Sentence not clear should be rewritten

Response> 

Authors agree that the sentence is not clear and is incomplete. Authors have removed the sentence and added a few lines dedicated to the reasons behind the occurrence of saline lands and their subsequent effects.

L27P2 "Monocots such as Zea..." please provide reference

Response >

Authors have provided the references for “Monocots such as Zea mays…..”  at the end of two sentences as “18 – 19” which discuss about the different categories of Si accumulators.

L4p3 "understand the researchers to focus on crops" sentence not clear, please rewrite.

Response>

Authors have rewritten the sentence to make it understandable. Please see the revised text

P3-P9 - description is focused on salinity stress, I suggest to combine it with response to Si

Response >

Here, authors have tried to focus only on “Role of osmoprotectant osmolytes during salinity stress”. Authors have combined this with response to Si in the later portion of the review under the heading “Si mediated biosynthesis of compatible solutes and phytohormone”

L25P9 "The concentration..." - not informative general statement, obvious. I think the reference [97] is not properly used in the case.

Response >

Authors have reframed the sentence using appropriate reference such the sentence is informative and not a generalized one.

L14-15P11 Word "similarly" do to describes results of both experiments.

Response >

Authors have replaced the word ‘similarly’ with ‘however’, so as to describe the difference in results and not similarity.

L19P11 - style, should be rewritten

Response >

Authors have rewritten the sentence as per the suggestion.

L10P12 should be "apoplast"

Response >

 Authors have made the correction as suggested.

L29P13 Do authors consider tomato as horticultural crops? If yes, the sentences should be rewritten.

Response >

Yes, authors do consider tomato as horticulture crop. Hence, the sentence has been rewritten accordingly. Please see the revised text

L31-33P13 Part of the text that should be rewritten. It is not necessary to provide aim of such short paragraph. "Under Si efficiency" - not clear.

Response >

Authors accept the suggestion given and therefore have removed the sentence as it was not clear in its context.

L12-13P14 Again very general sentence, please be more precise, focus on salinity.

Response >

Authors have rephrased the sentence such that it only focuses of salinity.

L18-20P13 The sentence is very similar to earlier sentence on the same study L38P13

Response >

Authors do not agree with the reviewer because both the sentences are completely different. Please see the revised text

26-28P13 the sentence on the impact of Si on Light harvesting and photoprotection is not supported by cited study [166].

Response >

Citation has been rewritten by the author to support the sentence. Please see the revised text.

L1P15 Should be [168]

Response >

Authors have made the correction in the revised manuscript

L10P15 RuBisCo was already described above.

Response >

Authors are trying to draw a parallel between RuBisCo activities in different plant species under salinity stress and supplemented with silicon. Since the results are concordant therefore authors have mentioned RuBisCo activity in both the paragraphs.

Conclusions are poorly written, as I mentioned above, problems and research gaps should be identified. 

Response >

Authors have rewritten the conclusions and research gaps at various places in the manuscript and a detailed conclusion is provided at the end of the manuscript.

I suggest to provide form of Si used in experiments whenever possible it may be useful information.

Response >

Authors have provided information about the form of Silicon used wherever it was available in the previous research work/literatures. Please see the revised text.

Reviewer 2 Report

The review ''Silicon in Horticultural Crops: Cross-talk, Signaling and Tolerance Mechanism under Salinity Stress'' by Al Murad et al., gives an extensive overview of the literature on salinity stress. The manuscript is well written and organised, however there are several typos to correct before publishing. Moreover, the references must be formatted (eg, abbreviations of journal headlines are sometimes punctuated and sometimes not), and I suggest to format also the name of cited plant species (some are given in vulgar, some in latin and some both ways). Following, specific comments are reported:

Pag1, line 7: Add a dot after ''Oman''

Pag2, line 9: Add ''of'' between ''form'' and ''two''

Pag4, line 8: It is not appropriate to start the  sentence with ''Figure 1''. Please

reformulate it.

Pag8, line 29: In the word ''stipulated'' the d is underlined, please correct it.

Pag10, line 21: Replace ''(OH) ,'' with ''(OH),''.

Pag 11, lines 1-2: Replace ''(POD) ,'' with ''(POD),'' and ''[113] .'' with ''[113].''.

Pag 11, line 38: Replace ''applied ,'' with ''applied,''.

Pag 12, lines 20,26: Replace ''vacuoles ,'' with ''vacuoles,'', ''[142] .'' with ''[142].'' and ''activity ,'' with ''activity,''.

Pag 13, lines 2,6,38: Replace ''stress .'' with ''stress.'',''Sorghum'' with ''sorghum'' and ''effectively .'' with ''effectively.''.

Pag 14, line 4,24: Avoid to start the sentence with ''17%'', replace ''[167].In'' with ''[167]. In''.

Pag 15, line 2,11: Replace ''245'' with words, replace ''Plant'' with ''plant''.

Pag 16: Check again all references, are not well formatted.

Author Response

Reviewer #2

The review ''Silicon in Horticultural Crops: Cross-talk, Signaling and Tolerance Mechanism under Salinity Stress'' by Al Murad et al., gives an extensive overview of the literature on salinity stress. The manuscript is well written and organised, however there are several typos to correct before publishing. Moreover, the references must be formatted (eg, abbreviations of journal headlines are sometimes punctuated and sometimes not), and I suggest to format also the name of cited plant species (some are given in vulgar, some in latin and some both ways). Following, specific comments are reported:

Response>

Authors have revised the whole manuscript as per reviewers suggestions and are highlighted in red text in the revised version of the MS.

Pag1, line 7: Add a dot after ''Oman''

Response >

Authors have added a dot after ''Oman'' as instructed.

Pag2, line 9: Add ''of'' between ''form'' and ''two''

Response >

Authors have added “of” between “form” and “two”. Please see the revised text

Pag4, line 8: It is not appropriate to start the sentence with ''Figure 1''. Please

reformulate it.

Response >

Authors have rewritten the sentence to remove “figure 1” from the starting of the sentence. Please see the revised text

Pag8, line 29: In the word ''stipulated'' the d is underlined, please correct it.

Response >

Authors have corrected the word as instructed. Please see the revised text

Pag10, line 21: Replace ''(OH) ,'' with ''(OH),''.

Response >

Authors have replaced the word as suggested in the revised text

Pag 11, lines 1-2: Replace ''(POD) ,'' with ''(POD),'' and ''[113] .'' with ''[113].''.

Response >

Authors have made the corrections as suggested in the revised text.

Pag 11, line 38: Replace ‘‘applied,'' with ''applied,''.

Response >

Authors have made the correction as suggested by the reviewer, please see the revised text.

Pag 12, lines 20,26: Replace ''vacuoles ,'' with ''vacuoles,'', ''[142] .'' with ''[142].'' and ''activity ,'' with ''activity,''.

Response >

The authors have replaced the words in the correct form as suggested. Please see the revised text.

Pag 13, lines 2,6,38: Replace ''stress .'' with ''stress.'',''Sorghum'' with ''sorghum'' and ''effectively .'' with ''effectively.''.

Response >

Authors have replaced the wrong form of the word with right form, as suggested.

Pag 14, line 4,24: Avoid to start the sentence with ''17%'', replace ''[167].In'' with ''[167]. In''.

Response >

Authors have formatted the sentence such that the sentence doesn’t start with “17%”.

Authors have also replaced words in the right form as suggested.

Pag 15, line 2,11: Replace ''245'' with words, replace ''Plant'' with ''plant''.

Response >

Authors have done the corrections as suggested. “245” was replaced with “About 245”.

Pag 16: Check again all references, are not well formatted.

Response >

Authors have checked the references and formatted them again as instructed. Besides, new references added were also included and formatted in the revised manuscript wherever necessary.

Round 2

Reviewer 1 Report

Authors answered most of the comments.